# Influences on policy-formulation, decision-making, organisation and management for maternal, newborn and child health in Bangladesh, Ethiopia, Malawi and Uganda: The roles and legitimacy of a multi-country network

Kohenour Akter[1]*, Yusra Ribhi Shawar[2,3], Anene Tesfa[4], Callie Daniels Howell[5], Gloria Seruwagi[6], Agnes Kyamulabi[6], Albert Dube[7], Geremew Gonfa[4], Kasonde Mwaba[5], QCN Evaluation Group[¶], Mary Kinney[8], Mike English[9], Jeremy Shiffman[2,3], Nehla Djellouli[5], Tim Colbourn[5]*

1 Perinatal Care Project, Diabetic Association of Bangladesh, Dhaka, Bangladesh, 2 Bloomberg School of Public Health, John Hopkins University, Baltimore, United States of America, 3 Paul H. Nitze School of Advanced International Studies, John Hopkins University, Washington, D.C., United States of America, 4 Ethiopian Public Health Institute, Addis Ababa, Ethiopia, 5 Institute for Global Health, University College London, London, United Kingdom, 6 School of Public Health, Makerere University, Kampala, Uganda, 7 Parent And Child Health Initiative PACHI, Lilongwe, Malawi, 8 School of Public Health, University of the Western Cape, Cape Town, South Africa, 9 Centre for Tropical Medicine and Global Health, University of Oxford, Oxford, United Kingdom

¶ Membership of the QCN Evaluation Group is listed in the Acknowledgments.
* t.colbourn@ucl.ac.uk (TC); mary05kh@gmail.com (KA)

## Abstract

The Network for Improving Quality of Care for Maternal, Newborn and Child Health (QCN) is intended to facilitate learning, action, leadership and accountability for improving quality of care in member countries. This requires legitimacy—a network's right to exert power within national contexts. This is reflected, for example, in a government's buy-in and perceived ownership of the work of the network. During 2019–2022 we conducted iterative rounds of stakeholder interviews, observations of meetings, document review, and hospital observations in Bangladesh, Ethiopia, Malawi, Uganda and at the global level. We developed a framework drawing on three models: Tallberg and Zurn which conceptualizes legitimacy of international organisations dependent on their features, the legitimation process and beliefs of audiences; Nasiritousi and Faber, which looks at legitimacy in terms of problem, purpose, procedure, and performance of institutions; Sanderink and Nasiritousi, to characterize networks in terms of political, normative and cognitive interactions. We used thematic analysis to characterize, compare and contrast institutional interactions in a cross-case synthesis to determine salient features. Political and normative interactions were favourable within and between countries and at global level since collective decisions, collaborative efforts, and commitment to QCN goals were observed at all levels. Sharing resources and common principles were not common between network countries, indicating limits of the network.

**Data Availability Statement:** The original qualitative data are not made available since some responses contain potentially identifiable information at multiple levels including for individuals, health facilities, and organisations; the Institutional Review Board conditions stipulated that participants will not be at risk of identification through their responses; participant consent forms did not include obtaining explicit consent regarding the possible use of anonymised data in the public domain via a data repository. Non-author contact information for enquiries about data access: please email dataprotection@ucl.ac.uk.

**Funding:** This work was funded by the Medical Research Council (MRC) Health Systems Research Initiative 5th call via grant MR/S013466/1 to TC at UCL Institute for Global Health, United Kingdom, YS and JS at Johns Hopkins University, United States of America, KA and AK at Diabetic Association of Bangladesh Perinatal Care Project, Bangladesh, CM at Parent and Child Health Initiative, Malawi, GS at Makerere University School of Public Health, Uganda, and ME at University of Oxford, United Kingdom; and by the Bill & Melinda Gates foundation via grant INV-007644 to TM at LSHTM, United Kingdom. The funders had no role in study design, data collection and analysis, decision to publish, or preparation of the manuscript.

**Competing interests:** The authors have declared that no competing interests exist.

Cognitive interactions—those related to information sharing and transfer of ideas—were more challenging, with the bi-directional transfer, synthesis and harmonization of concepts and methods, being largely absent among and within countries. These may be required for increasing government ownership of QCN work, the embeddedness of the network, and its legitimacy. While we find evidence supporting the legitimacy of QCN from the perspective of country governments, further work and time are required for governments to own and embed the work of QCN in routine care.

## Introduction

The Network for Improving Quality of Care for Maternal, Newborn and Child Health (QCN) [1] was created to reduce maternal, newborn and child health morbidity and mortality by improving quality of care. QCN was intended to facilitate learning, action, leadership and accountability for improving quality of care in member countries [2]. QCN comprises a global secretariat based at the World Health Organization and is led in each of the 11 member countries by directorates in Ministries of Health [3]. In each member country, QCN is made up of individuals representing a range of institutions including non-governmental organizations, professional associations, hospitals and health facilities, and district, regional and national ministries of health [3]. The decision to opt into QCN was largely made by a country's ministry of health in consultation with the network's global secretariat [3]. The participants in the network are motivated by improving quality of care–increasing effective coverage–of interventions to reduce maternal, neonatal and child health mortality in hospitals [3]. The roles and responsibilities of these participants in the network have included developing national road-maps toward improved quality of hospital-based care for mothers and newborns, developing and implementing initiatives to improve quality of care at individual facilities, and strengthening data collection and use on indicators related to quality of care at hospital, district and national levels [3, 4]. Our work on the emergence of QCN in four countries found that it emerged quickly and most robustly in Bangladesh, followed by Ethiopia, then Uganda, and Malawi [3]. Our work on the effectiveness of QCN found that "global and national leadership elements of QCN have been most effective to date, with action, learning and accountability more challenging, partner or donor dependent, remaining to be scaled-up, and pandemic-disrupted" [4].

For the network to work as intended the government of each member country must buy into, trust, and spend time and resources on network activities [2]. Each member country therefore must recognise the legitimacy of the network and take sufficient ownership of policy and management activities required to deliver the strategic objectives of the network [4]. In this paper, legitimacy is understood as 'the right to exert power'[5]. This right can be understood both in a normative and empirical sense. The former, from the perspective of democratic theory questions if an actor has a right to exert power, i.e., is QCN, which includes the global secretariat, as well as national and local actors, representative of constituent interests and/or historically effective in meeting those interests? The latter examines an actor's perceived right to exert power from the perspective of a particular actor—in our case in this paper the ministry of health department that leads the network in country. For example, do domestic actors perceive the engagement and influence of QCN—especially of those actors that are external to the country (i.e., global secretariat and representatives from other involved QCN countries)—to be permissible? However, democratic theorists, contend that the right to exert power is

contingent not just on performance—what they term output legitimacy—but also fair process, inclusive deliberation and transparency—or input legitimacy [6–8].

Social science scholarship identifies several characteristics that foster network legitimacy. For example, international studies scholarship on international organisations indicates that social trust, democratic organisation, and how these are influenced by prior communication and beliefs compatible with the mission and values of the initiative are likely to foster legitimacy; the absence of these makes an initiative or actors less likely to be seen as legitimate [9–12]. Other scholarship, from the perspective of expert stakeholders, find an organisation's performance, its purpose, and procedure to drive legitimacy [13]. Scholarship in sociology emphasizes consideration of institutional contexts and/or histories of the involved countries in considering an initiative's outcomes and perceived legitimacy [14, 15]. For example, Robinson (2017) illustrates how preceding experiences in family planning prefigured—and directly impacted the perceived legitimacy and nature of—national HIV prevention strategies in sub-Saharan Africa [16].

In this paper, we examine QCN's legitimacy—'right to exert power' [5]—in advancing policy and improving services from the particular perspective of national government departments that led QCN country teams—across four of the involved countries: Bangladesh, Ethiopia, Malawi, and Uganda. Specifically, we investigate QCN's legitimacy by analysing the nature of the interactions across global, national, and local network actors engaged in QCN. Consequent to QCN's legitimacy we also investigate the ownership and direction of strategies adopted in each country, and how embedded (integrated) the work of the network is in the health system (routine care) of each country [17]. Following Vanyoro et al [17], who define ownership of research, we define ownership of implementation of the QCN here as: "the process whereby co-production enables health system actors (from policymakers to service users) to determine and influence [implementation] agendas with direct engagement with the [implementation] process itself". We consider the context of each country in our investigations.

This paper on network legitimacy and ownership is part of a series of papers evaluating the QCN and complements our papers on network emergence [3] and network effectiveness [4] [S1 Text: 2-page summary explaining collection of QCN papers]. Following the emergence of the network at global and national levels [3] this paper looks at interactions between the institutions involved in each country, and the global level, which is key to understanding network effectiveness, as well as specific aspects of the work of the network such as innovation, sharing and learning and our stakeholder network analysis [18], looking at interactions between QCN actors from a quantitative perspective. Understanding the factors shaping legitimacy of QCN is important both for understanding the emergence and effectiveness of QCN, and for the success of future multilateral international efforts that bring governments and multiple stakeholders together to improve health systems and quality of care, and for work on other initiatives more broadly.

## Methods

QCN emerged during 2017–2019, involving 11 countries, and was disrupted by the COVID-19 pandemic [3]. Our study was carried out at national and local levels in Bangladesh, Ethiopia, Malawi and Uganda as well as at the global level of QCN. We chose these four countries as case studies as they represent a range of maternal, newborn and child health contexts and prior histories with quality improvement efforts.

Our qualitative study involved, over three years (2019–2022), an iterative series of interviews with key stakeholders, observations of meetings at local and national levels as well as

hospital observations, and a document review. At the initial stage, we selected a few stakeholders who attended the network's first global meeting and then followed a snowball strategy (asking respondents who else is involved in the network) to find key stakeholders at all levels of the health system in each of our case study countries. In total, we conducted 227 interviews across the four case study countries in two to four rounds of data collection. We also conducted 21 interviews with global level network members and stakeholders over two rounds. Interviews were mostly 45 to 60 minutes long. We conducted non-participant observations of multi-country meetings and key national-level and district level meetings in case-study countries. Activities at district level were also observed via visits to two best and two least performing QCN hospitals in each case study country in several iterative rounds. We further reviewed accessible published and unpublished documents and communications relating to the QCN at global level and at national and sub-national levels in the case study countries. These included strategy and management documents, operational plans, directives, formal minutes, and reports. We were able to access unpublished documents via WHO and Ministry of Health QCN contacts.

The iterative nature of our work, which included follow-up interviews of many respondents over several years during the evolution of QCN, with accordingly iteratively revised interview topic guides, enabled us to investigate how institutional interactions and consequent legitimacy and embeddedness of the network changed over time, up until 2022. The COVID-19 pandemic also disrupted our research, though like QCN itself, some work, such as interviews, moved online. Please see S2 Text [common methods document] for details of all data collection methods and how this study is linked to the wider evaluation of the QCN we undertook. Here we focus on the framework and theories we use and our analytical methods for this paper.

## Legitimacy framework

To guide our analysis, we developed a framework (Fig 1) drawing on three relevant models. First that by Tallberg and Zurn [9] which conceptualizes legitimacy of international organisations as being dependent on their features (authority, procedure, performance), the legitimation process (intensity, tone, narrative), and legitimacy beliefs of audiences (constituents and observers). Second, by Nasiritousi and Faber [13], which looks at legitimacy in terms of the focus of institutions on a problem, looking at purpose, procedure, and performance of institutions. We use this to consider how the history of work on quality improvement in each country by the institutions involved in QCN relates to observed legitimacy, ownership and embeddedness of QCN in the country. Third, a model developed by Sanderink and Nasiritousi [19] to characterize networks in terms of perceived institutional interactions. This divides institutional interactions into political, normative, cognitive, behavioural and 'impact level' interactions. We focused on three of these interactions, political, normative and cognitive (Fig 1) to investigate the legitimacy and ownership of the work of the network in each case study country, looking at the extent to which different organisations involved in the network work together across these three dimensions. Political interactions are those related to joint decision making and collaboration; normative interactions are those related to shared principles, norms and commitments; and cognitive interactions are those related to information sharing and transfer of ideas [19]. We do not focus on behavioural or impact level interactions as behavioural change and impact are difficult to measure and are concerned with network effectiveness, the subject of another of our papers [4].

Institutional interactions may also be shaped by power relations between institutions, which may be dependent on the capacity of each institution [20, 21], e.g., institutions with

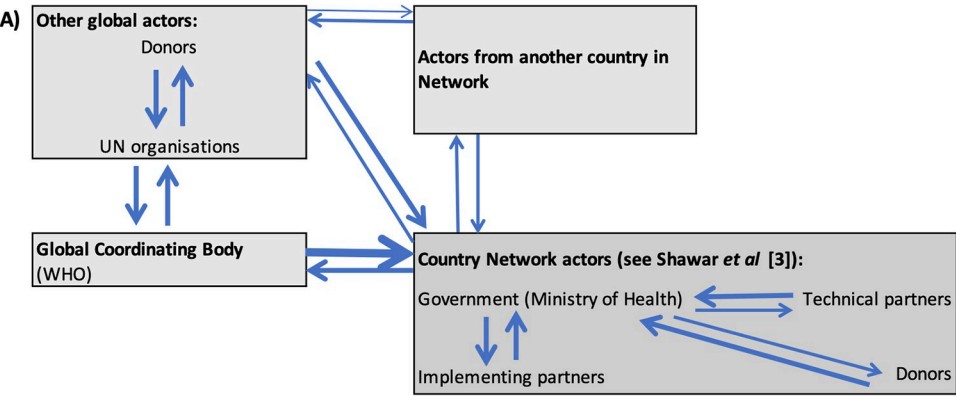

**A)**

Other global actors:
Donors

UN organisations

Actors from another country in Network

Global Coordinating Body (WHO)

Country Network actors (see Shawar *et al* [3]):
Government (Ministry of Health)
Technical partners
Implementing partners
Donors

Arrows represent Institutional Interactions, including:
Political Interactions    Normative Interactions    Cognitive Interactions

as defined by Sanderink & Nasiritousi [21]

**B)** **Strength of interactions between institutions (width of blue arrow in Figure 1 A) above) will depend on:**
- Capacity (see Wu *et al* [23], Tesfa *et al* [22]) e.g. Human resources available in country to draft policy and programmes, organise activties, analyse and evaluate processes and outcomes
- Alignment of goals between actors (**beliefs**, organisational **purpose**)
- Leverage (**authority**) via other agreements and influences, and **procedures**
- **Performance** of organisations
- Organisational culture
- Wider culture & Political stability

Nasiritousi & Faber [13]        Tallberg & Zurn [9]

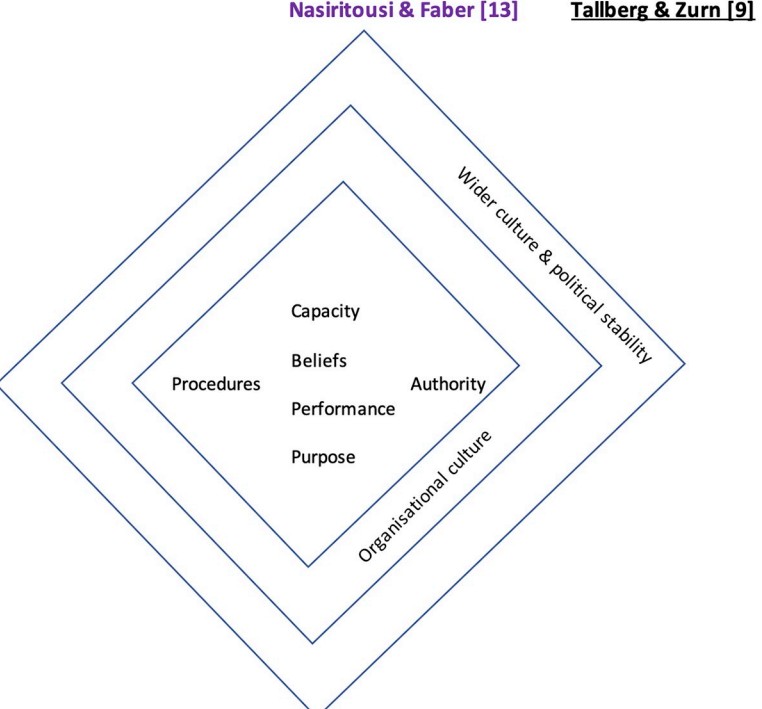

Wider culture & political stability

Capacity

Beliefs

Procedures          Authority

Performance

Purpose

Organisational culture

**Fig 1. Framework describing drivers of legitimacy and ownership of the work of quality of care network (QCN) from the perspective of national governments leading the work of QCN. A)** Depiction of interactions between institutions (actors) comprising the quality of care network (QCN) at global level (left) and national level (right). Interactions include political, normative and cognitive interactions between institutions as defined by Sanderink and

Nasiritousi [19]. The width of the arrows indicates the strength of the interaction and is illustrative only and analyzed qualitatively. The width of the arrow depends on the variables in part B of the figure. **B)** We draw on work on individual, organizational and system capacities [20, 21] and models by Nasiritousi and Faber [13] (purple text) and Tallberg and Zurn [9] (orange text) on sources of institutional legitimacy to describe factors influencing the strength of interactions between institutions. These are depicted as characteristics of organizations (center of the diamond), organizational culture (next layer of the diamond) and wider culture and political stability surrounding the organization (outer diamond).

greater capacity have more power to form policies and influence decisions and ways of working of other organisations. In our examination of interactions between institutions we also consider the nature of institutional agency and power in relation to structure, by considering distribution of financial and economic resources, organisational culture and ways of working, alignment of goals between actors, leverage via other agreements and influences and political stability (Fig 1).

We consider the history of quality improvement efforts in each country in relation to the formation of the network and wider context of maternal, newborn and child health programmes in detail in S3 Text common to all papers in our QCN evaluation series [common country context document]. In this paper we extract the most relevant aspects of this background information explaining the role of institutions involved in QCN at the beginning of our results section and follow with the results of our analysis of institutional interactions, legitimacy and ownership of the work of the network described above.

## Analysis

We used thematic analysis [22] of interview transcripts and process tracing [23] using interview data, review of key documents and observations of meetings to characterise political, normative and cognitive interactions between institutions involved in QCN in each country. We compared and contrasted these interactions in a cross-case synthesis to determine salient features of these institutional interactions in order to evaluate the legitimacy of the network and ownership and embeddedness of the work of the network in each country, and which contextual factors they depend on, to answer our research question.

## Ethics

Ethical approval was received from University College London Research Ethics Committee (ref: 3433/003); BADAS Ethical Review Committee (ref: BADAS-ERC/EC/19/00274), Ethiopian Public Health Institute Institutional Review Board (ref: EPHI-IRB-240-2020), National Health Sciences Research Committee in Malawi (ref: 19/03/2264) and Makerere University Institutional Review Board (ref: Protocol 869). The conduct of the evaluation was based on clear ethical standards which assured confidentiality, privacy, anonymity and informed consent. All respondents provided verbal or written informed consent. All respondents were informed of: (i) the purpose of the evaluation; (ii) their right to refuse to participate; and (iii) that their possible decision not to participate would not be held against them or affect their status in the network.

## Results

In examining QCN's legitimacy, we first summarize key contextual information concerning the roles of each network partner institution in each of the country cases; further details on QCN's emergence in each of the countries, as well as their histories concerning quality improvement and MNCH initiatives are provided in Shawar *et al* [3] and the S3 Text

[common country context document]. We then discuss the political, normative and cognitive interactions between involved actors.

## Bangladesh

Bangladesh's work on and government commitment to quality improvement long pre-dates the establishment of QCN. In terms of government commitment, the Quality Improvement Secretariat (QIS), established by the Ministry of Health and Family Welfare (MoH&FW) in January 2015, supports quality improvement (QI) initiatives across the country and strengthens and coordinates QI activities in the public and private health sector. QCN was integrated into QIS. In addition to QIS, there were several development partners that have long worked on quality improvement in the country, including WHO, UNICEF, USAID and Save the Children. UNICEF worked in partnership with the Bill and Melinda Gates Foundation and the Ministry of Health (MoH) since 2015 to demonstrate a model of quality of care to scale up at national level via its Every Mother, Every Newborn (EMEN) pilot project in Kurigram (one of the northern districts in Bangladesh) [24]. Save the Children is a key implementer of USAID's Mamoni Maternal, newborn and child health strengthening project (MNCSP), a flagship activity to support the Bangladeshi Maternal and Newborn Health program, started in 2018 [25]. These actors collectively engaged in the establishment and implementation of QCN activities since they had a long history of working on maternal and newborn health including quality of care [3]. They had implemented the QCN activities independently but with government support and connected to their previous work. Other actors, that did not appear to interact directly with QCN but contributed to QI implementation processes included UNFPA, the National Institute for Preventative and Social Medicine (NIPSOM) and district-level Civil Surgeons. NIPSOM is a government academic institution invited by UNICEF to play the role of national learning hub. NIPSOM was also working as implementing partner with UNICEF's support, and participated to train and coach facility health workers. URC from the global level also worked to train and coach health workers, especially during the initial stages of QCN, and sometimes via online sessions. The Civil Surgeon is the district head in health and implementing partners run the projects informing him of every detail. UNFPA works on Maternal and Perinatal Death Surveillance and Response (MPDSR) at the national level along with other partners, but is not part of QCN activities. Relevant departments of the Directorate General of Health Service (DGHS) and Directorate General of Family Planning (DGFPA) are also involved.

## Ethiopia

Ethiopia also had a history of MNCH and quality improvement initiatives prior to the introduction of QCN [2]. For example, in 2015, the government introduced the Health Sector Transformation Plan (HTSP), which sought to improve maternal and child health services. In 2016, the National Healthcare Quality Strategy (NHQS) was launched, followed by the establishment of quality units at federal, regional, district and facility levels. The country also had experience with similar network initiatives, including the Ethiopian Hospitals Alliance for Quality (EHAQ), initiated by MoH in 2012 [26]. In 2016, QCN was placed in the MoH, which played a leadership role in coordinating and providing technical support, coaching and mentoring for quality improvement activities. It established a technical working group (TWG) consisting of representatives of different partners and prepared a national roadmap called LALI (Leadership, Accountability, Learning, Implementation, alternatively used to LALA) [27] and identified learning facilities. QCN eventually became a country-led program, mainly coordinated by the MoH, with institutions, including international donors, and NGOs either

funding or providing technical support at national level (for example WHO, UNICEF, USAID and UNFPA) or implementing the program at the facility level (these include IHI, Transform Primary Health Care Unit (Transform PHCU), Transform Health in Developing Regions (Transform HDR), CHAI, and WHO). WHO played a vital role in initiating, directing and coordinating the implementation together with MoH. WHO also provided technical and financial support for some of the local facilities, up until 2021 when WHO ceased their QCN activities in Ethiopia. It also served as a link to the WHO headquarters and the QCN at the global level. UNICEF and UNFPA played the role of financial partners.

## Malawi

In Malawi, QCN built on previous government and partner efforts to reduce maternal and newborn mortality as part of the MDGs/SDGs, as well as efforts towards achieving universal health coverage (UHC) and work done on HIV trying to reduce mother-to-child transmission [28]. In November 2016, the government established the Quality Management Directorate (QMD) within the Ministry of Health, where QCN was placed. QMD aimed to contribute to improve health and client satisfaction via provision of quality health services and to drive the national agenda to improve quality and equity in the health sector in Malawi. After introducing the QCN to the MoH, the WHO assisted the Ministry in gathering key stakeholders, which formed a coordinating body, the Techincal Working Group (TWG) in charge of planning the implementation of QCN in the country. Other key stakeholders at the national level include the Reproductive Health Directorate (RHD) of the MoH, UNICEF, UNFPA and GIZ. RHD was a technical partner and worked with QMD in supporting and coordinating network efforts though they were less visible in QCN efforts over time. GIZ and UNICEF were playing the roles of implementation, technical and funding partners and also supported other community-based organizations (e.g. Society of Medical Doctors (SMD) and MaiKhanda) directly to implement QoC activities. UNFPA was providing technical assistance to develop policies and strategies, providing funding to RHD and QMD, and playing the role of an implementing partner. Other stakeholders who also played substantial roles at the national level include PACHA (Paediatrics and Child Health Association), NEST 360 (NEST 360 is an international alliance of clinical, technical and public health experts from 17 leading institutions, governments and organizations), ONSE (Organized Network of Services for Everyone's), and Cowater where PACHA (Pediatric and Child Health Association) was playing both the role of technical and implementing partner and other organizations were working in implementation. Jhpiego, CHAI (Clinton Health Access Initiative), Save the Children and EGPAF (Elizabeth Glaser Pediatric AIDS Foundation) assisted to review the roadmap.

## Uganda

QCN in Uganda built upon a long history of QI initiatives that remain ongoing, particularly those focused on HIV, reproductive health, and malaria. Previous QI initiatives in Uganda were Yellow Star and using the 5S's (sort, set, shine, standardize, and sustain) approach in HIV, TB and malaria, which established QI teams at each level of the health system as well as specific standards, indicators and databases. Uganda's commitment to improving MNCH was exhibited in its 2013 RMNCAH Sharpened Plan for Uganda, a national RMNCAH policy which set out to address existing bottlenecks to reduce maternal, newborn and child morbidity and mortality [29]; the updated plan in 2021, sought to especially focus on quality of care.

QCN was originally co-led by the government's Quality Assurance Department and MCH department. QCN only began to flourish in 2019 after the renaming of the Quality Assurance Department to the Standards Compliance Accreditation Patient Protection (SCAPP)

department under the Directorate of Governance and Regulation, and assigning it sole oversight and appointing a focal person for QCN, who brought more partners and funding on board. In this new arrangement, SCAPP would still work with the other departments but took responsibility for Network activities. In line with the country's decentralised health system regional quality improvement teams (QIT) were established, which aimed to lead and support district and health facility QITs. Several implementing partners played crucial roles in QCN in Uganda. The WHO introduced the QCN to the MoH and helped gather key stakeholders to form a TWG. Other key stakeholders in the country at the national level, included USAID, UNICEF, UNFPA. USAID worked through their partner organisations in Uganda: URC (previously) through Applying Science to Strengthen and Improve Systems (ASSIST) RHITES (North for Acholi region and East central for Busoga region), RHITES-Southwest/EGPAF, Save the Children and Family Health International (FHI) (specifically under the MNCH/N activity). UNICEF and UNFPA also supported other community-based organisations (CBOs) to directly implement QoC activities, e.g., The Association of Volunteers in International Service (AVSI) and International Training in Health (IntraHealth). Over the course of 2021, UNFPA increased its involvement with QCN and decided to formally enter the network rather than mirror its work independently. Other stakeholders who also played substantial roles at the national level were the Makerere University School of Public Health as the designated learning partner on the Network, CHAI, Jhpiego, and Ugandan professional associations including the Ugandan Paediatric Association and the Uganda Private Midwives Association. Another stakeholder playing a large role in the QCN in Uganda indirectly, was the World Bank through its Global Financing Facility (GFF), which funded MOH's Uganda Reproductive Maternal Child Health Services Improvement Project (URMCHIP) project, though it was not a direct QCN partner. Most involved partners requested to join the network themselves and were already Ministry partners on Sexual Reproductive, Maternal, Newborn, Child and Adolescent Health (SRMNCAH) issues.

## QCN legitimacy within and across countries

QCN's legitimacy is understood to be comprised of several types of interactions: political, normative, and cognitive. We first present our findings on political interactions, then normative interactions, and finally cognitive interactions, bearing in mind how the extent of the institutional interactions will depend on the capacity, beliefs, performance, purpose, procedures and authority of each organisation, it's wider organisational culture, and wider culture and political stability of the country (Fig 1). The presence (X) or absence (blank) of these types of interactions between the major institutions mentioned above, is summarized in Table 1, both for institutions within each country, and from the country to other countries in the network or the global level. All of these interactions drive (Fig 1) and reflect legitimacy of the QCN in each country, whilst for government ownership and embeddedness of the QCN in each country we are specifically interested in sharing of resources and transfer of concepts from the national level to other network countries or the global level of the network.

**Political interaction.** The political institutional interactions we examined included those related to collective decisions, collaborative efforts, and resource sharing among key network partners across local, national and global levels. Political interactions between QCN institutions appeared strong in all four case study countries.

*Collective decisions.* Collective decisions, evidenced by a Memorandum of Understanding (MoU) or joint statement for example, should be artifacts of any multi-stakeholder network [19] and both bolster and reflect legitimacy of the network. Collective decisions were observed at the global level as the network emerged, where all partners had an agreement with the WHO-based QCN secretariat regarding the formation of the network. This suggests QCN

**Table 1. Types of institutional interactions observed in case study countries.**

| Interaction type | Bangladesh | | Ethiopia | | Malawi | | Uganda | |
|---|---|---|---|---|---|---|---|---|
| | Within country | Bangladesh to^ other network countries or Global level | Within country | Ethiopia to^ other network countries or Global level | Within country | Malawi to^ other network countries or Global level | Within country | Uganda to^ other network countries or Global level |
| **Political interactions** | | | | | | | | |
| - Collective decisions | X | X | X | X | X | X | X | X |
| - Collaborative efforts | X | X | X | X | X | X | X | X |
| - Resource Sharing^ | X | | X | | X | | X | |
| **Normative interactions** | | | | | | | | |
| - Shared commitments | X | X | | X | | X | X | |
| - Shared norms | X | X | | | | | X | |
| - Common principles | X | X | | X | | X | X | X |
| **Cognitive interactions** | | | | | | | | |
| - Exchange of information | X | X | X | | X | X | X | |
| - Transfer of concepts and methods^ | | | | | | | | |

^ For our analysis of government ownership and embeddedness of the QCN in each country we are specifically interested in sharing of resources and transfer of concepts from the national level to other network countries or the global level of the network

legitimacy is linked to legitimacy of the WHO. One of the global respondents depicted the network as an agreement to not duplicate resources and properly utilize existing resources.

At the national level, partners of all case countries working for QCN implementation (primarily setting up agreements and monitoring), including those in the private sector, have contracts or longstanding bilateral arrangements with the MoH and with each other. The prior performance of partner organisations in each country–their history of contributing to quality improvement efforts–influenced their contribution to collective decisions. In addition, the formation or reformation of TWG at the national level was an example of collective decision making. Development of QCN roadmap, preparation of national strategy and establishment of standards–were the reflection of joint statements or collective decisions.

Implementing partners in each country usually co-produced knowledge, guidance and gave statements on key issues with the MoH but were keen for the MoH to be seen as taking the lead and were in support roles of the ministry's strategic and operational direction.

*"In new districts, at first we have one to one interaction with leadership where we give some overview. After that, we organized an inception meeting, all of the leaders attended that meeting, and through that process, we make them oriented as well as engaged with our activities."* (Technical and Implementing Partner–National level–Bangladesh Round 3)

In Bangladesh, QIS and other partner organizations attended the follow-up meeting with WHO QCN secretariat together and sent a narrative report and working plan to the WHO

QCN secretariat, which was a good example of joint work. Another example of a collective decision was observed in Ethiopia in identifying partners and selecting learning sites by MoH and WHO in country, at the initial stage of QCN.

Autonomous decision-making persists along with coordination and collective decisions. For instance, in Uganda, partners took the autonomous decision during the selection of sites for scaling up. Bangladesh also experienced independent decision-making (e.g. selecting scaling up areas) and influence on MoH (e.g. joining the network, running capacity building activities by NIPSOM) in decision-making by the partners. The partners in Malawi also seem to have a lot of autonomy concerning decisions of which activities they will support and where. Similarly, Ethiopia also has the experience of autonomous decision-making on site selection since the selection of facilities was made based on partners existing support or pre-existing support by another project.

*Collaborative efforts*. Collaborative efforts include co-organizing events, co-coordinating activities, or co-authoring publications. Partners across global, national and local levels, and especially key national level partners in all four case study countries displayed strong collaborations. Cohesive participation in developing the forthcoming National Health Quality Strategy (NHQS) by all key partners at the national level in Bangladesh was observed for instance, though USAID was steering the strategy and communicating with different directorates as well as the WHO global network.

Developing QCN roadmaps in Ethiopia, Malawi and Uganda were the result of collaboration between QCN partners and in Uganda, this collaboration at the national level increased after MoH's leadership transition which also made clear roles and responsibilities of the different units. Many respondents expressed their realization of the importance and benefit of QCN setting-up opportunities for collaboration.

Co-organizing different events, workshops and training were common across all study countries though the events were led by different partners depending on the topics. For example, in Bangladesh, national events were mostly led by government; they led in agenda setting and decision-making, and development partners supported technically and/or financially, so they could push the government to organize such types of collaborative events. Such financial and technical support to proceedings provided development partners authority to strategically influence QCN, though, as in Bangladesh, government often set the agenda and made the decisions on what was implemented and by which organizations. This was not always the case though. Two perspectives were observed regarding funding and decision-making in Malawi—influence of development partners in implementation and conditional funding, which is experienced through UNICEF, and reliance on partners for direction.

> "…..*we are hoping that partners like UNICEF*, *GIZ would come and say; 'okay*, *what will be our direction*?' *This is because other than WHO*, *we need to engage other partners*" (Government - National level – Malawi Round 3)

*Sharing resources*. At the global level, BMGF and USAID were the primary funding partners. BMGF primarily supported through funding the WHO-based QCN secretariat and UNICEF for national implementation. Later, BMGF did not fund network activities beyond the global secretariat which shrunk the implementation activities in UNICEF funded countries and spaces though they overcame this quickly through alternative funds. For instance, in Bangladesh, BMGF was the primary funder for the Kurigram project, while they also worked in 5–6 other districts with funding from Global Affairs Canada and the UN Emergency Fund for the Rohingya in Cox's Bazaar. USAID funded QI efforts in-country via the MoMENTUM

award, a global initiative covering 30 countries including Bangladesh and Malawi (but not Ethiopia as MoMENTUM award was not running there and Uganda because of their own system) that will continue some of the QCN activities after QCN funding stops in 2023 [30], with funding going directly to implementing partners rather than MoHs. Involvement of most funding partners in all four case countries was either via the continuation of previous QI efforts or MNCH activities, or via alignment with QCN activities (e.g., Mamoni project in Bangladesh). QI was also incorporated into pre-existing work, for example, World Bank/GFF supported funding for QI activities in Uganda, but they didn't really align themselves with QCN activities.

Funding or sharing resources at the national and local levels is catalytic to quality improvement activities. In addition to receiving direct budget support from donors, it was common for partners in all four case study countries to mobilise their own resources, though the amount of resources varied across countries. Since there was no specific budget for QCN in Ethiopia, partners were using their own budgets earmarked for other similar QI activities to prepare learning sessions and support facilities. Big partners like USAID, UNICEF, WHO, Save the Children also provided human resources support through providing direct funding to the government or through implementing partners or by themselves at the national and/or facility level in all countries.

> *"We provided funding to the government directly to hire additional human resources like officers for/located at the regional hospitals"*. (Technical and Implementing partner–National level—Uganda Round 1)

Aside from human resources support, partners also provided logistics and financial support and additionally, in some cases, support in the reconstruction and renovation of different facilities for MNH services across our four case countries. Pooling resources depended on necessity and on the ministry's request and/or facility manager's request. For example, in Ethiopia, rather than focusing solely on quality, all activities were considered such as project design, implementation planning, and so on and when a need developed, such as when the government requests assistance or when gaps exist, donors such as UNICEF helped to fill those gaps, either in kind or cash. In Bangladesh and Malawi, partners also directly provided resources (e.g., equipment) to learning sites. In Uganda, partners pool their resources at the national level for QCN activities but previous funding experiences and prevailing implementing partner working arrangements with MOH, including tight timelines and targets, led funders to prefer directly funding their own QI initiatives whilst ensuring alignment and reporting to MOH priorities. This decision was taken by most funding partners, with the agreement of MoH.

Domestic resources have been used for QCN activities in Ethiopia and Bangladesh in MoH or development partner's initiatives. In Ethiopia, MoH eventually shared some budget with the regions. However, having financial autonomy, some of the regions used the money for other purposes. On the other hand, one partner in Bangladesh succeeded to convince the National Institute of Local Government (NILG) to properly utilize their budget for MNH. Our meeting observation and interview showed that Save the Children succeeded to do it by engaging NILG in different events, like, advocacy meetings, establishing functional linkage between health and family planning departments and NILG, engaging them in data-driven decentralized planning and regular communication and follow-up. This helped to develop ownership by local government and contributed to sustainability. Such type of devolved funding was not observed in Malawi or Uganda. This suggests that the organizational and political culture of the MoH–their willingness to allow devolution or flexibility of funding–influences allocation

of resources for QCN activities and reflects the extent to which the local level of QCN implementation was seen as legitimate by governments in each country.

However, our interviews and observations indicate that resource allocation was not sufficient in most facilities to run QCN activities smoothly. In Uganda, there was a common theme that the global level of QCN did not consider the physical constraints or level of rapid resourcing needed from the government to be able to achieve the goals and meet the standards, particularly around experience of care. A similar claim regarding organizational and structural capacity, came from respondents in Bangladesh.

**Normative interaction.** The commitment, norms and principles of all QCN actors of all countries overlapped with national goals and were consistent with previous works that have been discussed in the background section. However, all agreed that network activities gave them more impetus to act and be accountable at global level. All the actors desired synergy though they didn't always experience synergy in terms of joint working on implementation and achievement of goals.

The governments of participating countries were leading the QCN through their commitment for improving quality of care and adaptation of the Network's strategic objectives to their country contexts through growing the partnership with the different organizations. All case study countries also adopted and adapted the QoC standards considering their own country context, led by MoH or the responsible department of MoH [4]. Each country had exposure to QI activities through partners and mostly previous partners were working on network implementation. This was supposed to be beneficial, but this was not always the case. For instance, in Uganda, new coordination of actors was required as initial lack of clear coordination led to many participants reporting a lack of awareness or sense of cohesion. While participants at the national level were aware of and unified behind the QCN's goals, there was often a sense that each actor was continuing to operate in its predetermined silo at sub-national level. Another challenge in design for Uganda's in-country approach was that facility-level stakeholders, especially frontline workers were not fully oriented to the QCN separately; but rather some network activities were mainstreamed into other existing standard operating procedures, practice guidance or QI initiatives implemented by other implementing partners. A lack of a standardized implementation plan, including a clear set of timed targets, made it difficult to ensure alignment and cohesion around QI at all levels of the network in Uganda. One MoH participant reflected that having clear, time-bound and measurable commitments would have increased motivation and momentum by encouraging accountability. However, this situation was not static: our last round of data indicated better coordination and leadership from the centre.

Ethiopia experienced similar challenges throughout the entire implementation period. At the start of QCN, most of the actors in Ethiopia perceived network activities as WHO's work until the MoH announced it was their flagship initiative. In Ethiopia there was some disconnect between the federal and regional levels. The federal level and regional level blamed each other. The regional level complained that the central (federal) level didn't share anything or give clear direction, nor assign them with responsibilities properly, and that made them see the work as the federal level's project. On the contrary, the federal government was complaining about the regional level's lack of commitment whilst the regional level associated this with lack of capacity. In addition, regional level informants also mentioned they had a minimal sense of ownership.

> ". . ..*the MoH announced the QCN program in their own; they simply ask us to send them one or two participants in a meeting.*" (Government-Regional level-Ethiopia Round 1)

In the initial phases of QCN emergence, lack of coordination was also observed between the quality improvement and MCH departments of MOH. However, this was to improve later after a clear allocation of roles and responsibilities was undertaken. Lack of clear direction to the sub-national level was observed as a reason for slow inception of activities at local level in Malawi. Less concentration and poor political commitment of central level gradually made the RHD less active. Besides this, involvement of numerous partners, and persistence of discrepancies between the partners' and local objectives, created difficulties to coordinate and consolidate various efforts.

Evidence showed that implementation activities in Bangladesh were mostly 'DP [Development Partner] centric' despite the government chairing the QCN. Like the other three case study countries, awareness was mostly at central level, and the network was not known to most of the sub-national level actors who perceived the activities as partner's work. This perception was, however, transformed progressively though not entirely, through meetings, workshops, and training organized by implementing partners, and when health workers at facilities started to comprehend the benefits.

The above findings indicate that the beliefs of network actors about the purpose of QCN influence its legitimacy. This is exemplified at the local level, where the network was less well known by many health workers, and consequently had less authority.

**Cognitive interaction.** It was expected in the network that all pathfinder countries will be willing to transparently share data within the network, have a desire to learn and develop, and that the international actors and countries will join to learn from one another. Good synergy was expected, but rarely materialized beyond sharing information through implementing-partner-led efforts. This may be due to lack of government capacity to assemble, manage, analyse and share information and adapt programming quickly in response, organizational cultures unused to such dynamic network-dependent decision-making, or both. Overall, the lack of transfer of ideas and concepts between institutions and countries involved in QCN illustrated limits to the network, and the embedding of QCN work in government health systems.

*Exchange of information.* Learning and sharing occurred between and within countries and at various levels: global, regional, national, sub-national, district, sub-district and facility level. Here, sub-national, district, and sub-district levels are identified as local level (Fig 2). To continue the learning within country, district level learning networks were established in Bangladesh (in 2020) and Ethiopia (in 2019) [2]. A national level learning hub was also established in Bangladesh. QCN had a learning platform at global level and other methods of learning and sharing included: i) regular calls between partner countries and the QCN Technical Working Group (TWG) where countries share implementation progress and challenges, ii) a topical webinar series co-organized with partners, with a focus on sharing national level experience and know-how, and iii) in-person global level meetings where all global partners and network countries send delegations of eight to ten people.

Exchange of information between countries was executed by all case study countries through regular calls and international meetings. From Bangladesh, TWG representatives attend the meetings. The QCN secretariat usually communicated with the focal person of development partners, and they coordinated and joined meetings along with the MoH. The MoH and development partners then shared updates on implementation activities in country with the QCN secretariat. In Uganda, the Ministry of Health and the WHO country office had weekly calls with the QCN global leadership and participated in network meetings, and their interaction increased and improved over our study period.

Of our four case study countries, Bangladesh and Malawi participated and shared experiences in different webinars initiated by the WHO QCN secretariat. MoH of Ethiopia and Uganda were initially reluctant to share at the global level though they agreed subsequently. In

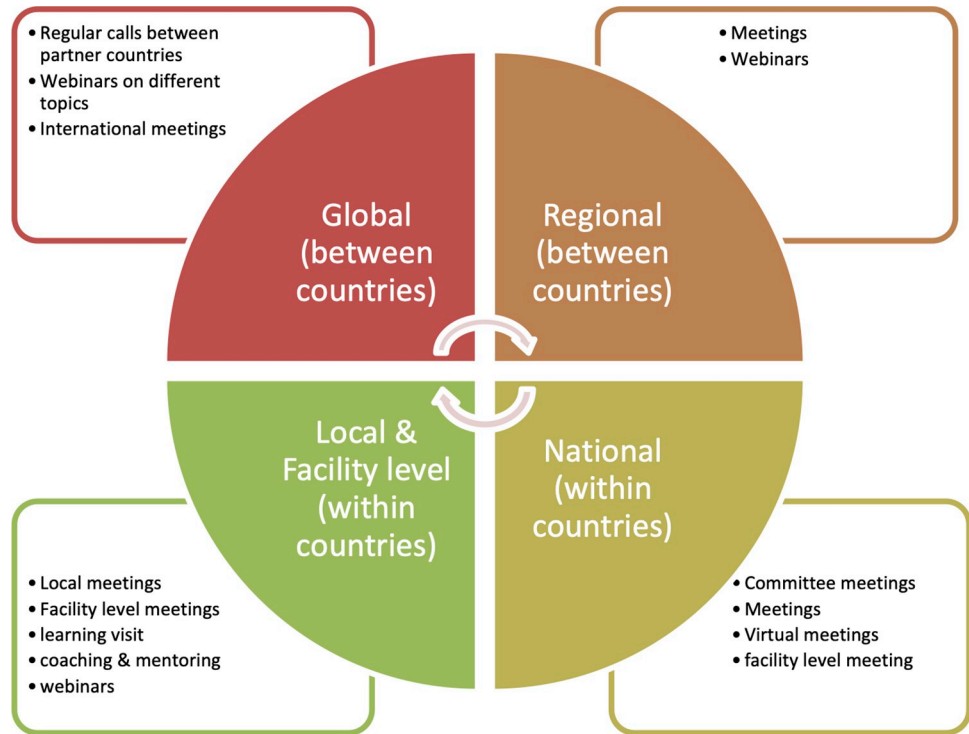

**Fig 2. QCN learning and sharing methods at global, regional, national and local levels.**

Bangladesh, development partners mostly coordinated it and included participants from local level also identified by them, with the consensus of the MoH. However, the webinars were mostly "DP [Development partner] centric" as country leadership didn't have time to engage with them. A national respondent in Bangladesh noted:

> *"Yeah! It's good but the government people don't have the time to attend/participate in this meeting. They don't want to talk, understand on this issue. So I think these are DP centric."* (Technical and Donor partner-National level-Bangladesh Round 4)

Though the learning platform is appreciated by most stakeholders, global participants mostly benefited and few stakeholders at national or local level attended.

All the partners including MoH, attended the global meetings, held during 2017–2019 before the COVID-19 pandemic, and shared experiences. Key development partners of Bangladesh also attended other global or regional meetings and webinars to share their methods, experiences and learning. Uganda also shared with neighbouring countries at the initial stage of QCN in their country.

Information exchange within countries was experienced by all countries in diverse ways and at both national and local levels. However, these meetings were mostly initiated and financed by implementing or technical partners, and dependent on their support. For example, facility level meetings in Malawi were initiated and financed by implementing partners, so stopped functioning for lack of resources when partners stopped supporting them. Shortage of budget and mistrust between the ministry and the regions affected regional level meetings in Ethiopia. In all four countries, the MoH also conducted national level sharing and learning meetings including large collaborative learning sessions, and implementing partners

undertook district collaborative learning sessions and facility-based learning visits to learning facilities within and between districts.

QCN was structured in Bangladesh and Malawi using already established national, subnational and facility level committees. Many of these were inactive though and started to be functional (mostly at the facility level) due to support and continuous focus by implementing partners.

*Transfer of concepts and methods.* Some respondents talked about adapting lessons from Bangladesh by other countries, though there was no evidence of transfer of concepts or methods by the study sites, i.e., neither the case study countries adopted learning from other countries nor other countries adopted learning from these countries.

Within countries, no one discussed implementation or adaptation of learning from other partner's quality improvement methods; rather all the partners' implemented QCN activities following their own approaches, and often in separate geographical silos. In Uganda, different partners operated in different regions and used different specific tools dictated by their different funding mechanisms. In Malawi, partners continued to support the kind of activities they had been doing before the launch of the QCN, working in similar areas. In Ethiopia, all partners used their own approaches, for example, one partner provided coaching every month, and another provided district-based coaching every quarter. Another one went to the districts from the center.

"*Partners have their own interests; they all have different approaches that they follow. For example, we say learning collaborative should be prepared in three months. Some do it within six months. Some conduct coaching every month, the others do it quarterly. Therefore, it lacks uniformity.*" (Government—National level—Ethiopia Round 1)

The two key partners in Bangladesh were following two different approaches. However, at the initial stage of the study, one stakeholder of one development partner (UNICEF), mentioned about a cross learning process, where they have learned 5s-CQI-TQM from the other partner (Save the Children)'s previous planned piloting area which was originally implemented by JICA Bangladesh. Similarly, the USAID team visited Kurigram as Kurigram was already established as a model district. A national respondent from Bangladesh noted:

"*Many [stakeholders] from different districts came to Kurigram including the whole USAID team. . .. . ..they visited Kurigram before they started the Project. They observed the measurement system including other good system*" (Technical, Implementing and Donor partner— National level—Bangladesh Round 1)

However, as our study progressed, no stakeholder talked about this type of visiting or learning. This may be due to a lack of embedding of such cross-learning in organizational cultures of institutions involved in QCN including the units of government ministries of health leading QCN in each country. Political instability, e.g., in Ethiopia, and the COVID-19 pandemic–in all countries–may also have contributed to QCN being unable to achieve significant cross-learning.

## Discussion

We found political interactions to be strong in all four case study countries supporting the legitimacy of QCN. In particular, collective actions and collaborative efforts were present both within countries and between countries, and between countries and the global level (Table 1). Resource sharing between QCN stakeholder institutions was also found within all four case

study countries, though not between countries. Nonetheless, the dependency on development partners and donors for resources limits the reach and depth of the network when their support is withdrawn and may also impinge on government authority to lead QCN-related work. In Bangladesh, the situation may be different as the next operational plan for quality should come with a separate government budget. Normative interactions including shared commitments and norms, and common principles were also observed in each of the case study countries, apart from shared commitments in Ethiopia, where there was some tension between the central (federal) and regional levels. Normative interactions extended between network countries and between network countries and the global level as shared commitment to achieving the goals of QCN and commitment to the WHO quality of care standards used by QCN. This built on shared commitments to global goals on maternal, newborn and child health over the last two decades and the legitimacy of WHO as a co-ordinating, technical and normative body driving, underpinning, and representing the global maternal, newborn and child health agenda.

In terms of cognitive interactions there was exchange of information within countries and between them, though bi-directional transfer of concepts and methods was more challenging and was generally absent within countries and between countries. Implementing partners typically implemented activities separately–using different concepts and methods, and in different geographical areas. This indicated the limits of government ownership of the QCN work and embeddedness of the network in that governments were not able to direct the work of the network to be cohesive. Specific activities were often determined by partners, differently in different areas dependent on which partner was operating where. Methods and concepts were not harmonised or synthesised and programmes of work often remained disparate and unconnected, despite the collaborative nature of the network and shared commitments indicated by the positive political and normative interactions. This lack of harmonisation and bi-directional transfer of concepts and methods may reflect lack of institutional capacity for this, or organisational culture not adapting to such new, networked, ways of operating. In many cases previous quality improvement efforts in the country, or district, and the partners that implemented them, shaped the specific work on quality for QCN.

Leadership at multiple levels to motivate individuals and to drive systems and policy and coordinate partner actions was identified by respondents as one of the core themes to drive the network [20]. QCN was frequently said to align well with government policy and to promote partner alignment across all network countries [20]. The leadership of QCN purposively identified and engaged NGOs that work on quality of care in each country, and globally. Having access to adequate physical resources, financing health care, and managing disruptive events all emerged as key drivers to network functionality [20].

Though the network aimed to link actors at facility level together and to those at district, regional and national level, the periphery of the network was found to be weaker, have less power, and be less networked [18] and coordinated than the central level. Our stakeholder network analysis found QCN to be a multi-hub network with less connections between actors at the periphery, and most connections between the centre and the periphery [18]. We also found the online learning platform to be predominantly used by global stakeholders rather than those at the periphery of QCN though those at district and facility levels were occasionally involved in collaborative learning sessions and learning visits. Actors at the periphery have limited power to change or improve local systems that are dependent on the central level (e.g., provision of human resources, procurement). The network did facilitate sharing of resources between partners and providers at local level though in some cases. However, state (MoH) actors at local and national levels lacked power to coordinate or pool such inputs. Together, these findings suggest that whilst the network was strong at global and national levels and

useful for advocacy and sustaining the policy-profile of QCN objectives it did not often extend to influence day to day changes in practice at facility level [4]. Organisational culture, and beliefs of network actors about the purpose of QCN may also have influenced its legitimacy, and consequent reach, at the local level [20].

We found the presence or absence of political, normative and cognitive interactions and resultant relatively high legitimacy of QCN and relatively low ownership and embeddedness of the work by governments to be similar across Bangladesh, Ethiopia, Malawi and Uganda. There were a few notable differences though including history of quality improvement in maternal, newborn and child health [3], different roles of different partners, and learning and sharing at the sub-national level. For example, the MoH in Ethiopia played an apparently leading role to run QCN activities and took the full leadership role since 2021 when WHO in country shrunk their activities in Ethiopia. Learning and sharing at sub-national level was experienced in Bangladesh more frequently than the other three countries.

Prior work has looked at legitimacy of agenda setting and prioritising specific issues in global health, for example, non-communicable diseases [31]. Prior work has also looked at legitimacy of specific organisations working in global health, for example the World Health Organisation [32, 33], or Bill and Melinda Gates Foundation [34], and has looked at the power relations involved [5]. In this study we examine the legitimacy of an implementation-focused network (QCN). Drawing on prior work described in our methods section [9, 13, 19, 21], we developed a framework to look at network legitimacy in terms of political, normative, and cognitive interactions between institutions involved in the network and determinants of the presence and strength of such interactions. We hope this framework may be useful in characterising the legitimacy of other implementation focused networks and the institutional interactions involved. The related concept of alignment may also be useful to consider going forwards as it has much in common with the concept of legitimacy. As described in a scoping review by Lundmark and colleagues [35], alignment has both structural (aligned plans and organisational structures) and social (cognitive, emotional and behaviour alignment of actors) dimensions and can be thought of as the process of creating a fit between inner and outer contexts of a system. Strategies to improve alignment include those pertaining to design and preparation, contextualisation, communication, motivation and evaluation of implementation efforts [35]. QCN has had some success so far in most of these areas [4, 30].

Recent work to develop a common understanding of networks of care reflects our findings by highlighting the importance of agreement, purposeful arrangements, buy-in and trusting relationships as enabling factors [36]. When assessing the results of applying their model to consider the effectiveness of multi-stakeholder partnerships for renewable energy, Sanderink and Nasiritousi found that sharing of procedural information and coordination mechanisms were most fruitful, though care was needed to ensure such interactions didn't harm the autonomy or efficiency of multi-stakeholder partnerships [19]. Our separate investigation of the effectiveness of QCN [4], and investigation of the legitimacy and embeddedness of QCN in this paper, reflects this: we found leadership and coordination aspects of QCN to be particularly strong and effective [4], whilst in this paper we find government autonomy is needed to embed and sustain the work of QCN to improve quality of care.

Key strengths of our study are the longitudinal iterative nature of the data collection over three years, the inclusion of four diverse case study countries and the global level of QCN, triangulation and synthesis of information between multiple methods including interviews, observations and document review, and use of recently developed models that specifically consider drivers of legitimacy and different types of institutional interactions necessary for legitimacy and ownership of the work of QCN by country governments. Our study is therefore robust, though key limitations remain. Not all instances of absence of a particular type of interaction, e.g., absence of sharing

resources between countries, was stated or corroborated by a wide range of respondents. Therefore, our findings, whilst likely to be broadly true, may lack some precision. We did not find evidence of interactions that led to less legitimacy of the network, hence our focus on the presence or absence of interactions that lead to greater legitimacy of the network. We did not find large differences in perceived legitimacy and government ownership of QCN work between our four case study countries despite large divergence in both the extent to which QCN emerged in each of them [3] and in how effective QCN was in each of them [4]. It may be that the relatively high legitimacy and low ownership of the work of QCN that we found across Bangladesh, Ethiopia, Malawi and Uganda is common to the other seven countries in QCN, or it may be that other countries had lower or higher legitimacy or ownership from the perspectives of their governments. Further research that more explicitly examines the linkages between a network's emergence, effectiveness, and perceived legitimacy would be useful.

The findings from this paper are useful as context for our assessments of the effectiveness of the network in delivering interventions and changing processes of care [4] and in understanding how it operates. In addition, the findings support and build on scholarship in international studies and sociology, which find legitimacy to be influenced by a network's mission and value compatibility with local contexts, as well as a country's history [13–16]. Further research looking at legitimacy and ownership of the work of QCN, and networks of care more broadly [30], at district, health facility and community levels within countries will be useful to deepen understanding of what drives networks and how best to embed their work into routine systems and sustain them.

## Conclusion

We found QCN legitimacy to be supported by shared commitments, norms and principles, developed from a long history of commitments to maternal, newborn and child health held in common, collective decision making, and collaborative activities. Encouraging pooling of resources and empowering peripheral levels may increase perceived legitimacy, and reach, of the network. Further work is required to develop government ownership of the work of QCN and embed it into routine systems. Enabling governments to synthesise and harmonise often diverse methods and approaches to quality improvement brought by different partner organisations, often working in different geographical areas, may be the key to this. Via such work governments may be able to embed processes to ensure higher quality of care for mothers, newborns and children across national, district and local health systems.

## Supporting information

**S1 Checklist. Inclusivity in global research checklist.**
(DOCX)

**S1 Text. PLOS GLOBAL HEALTH QCN evaluation collection 2-page summary.**
(DOCX)

**S2 Text. QCN papers common methods section.**
(DOCX)

**S3 Text. QCN papers common country context.**
(DOCX)

## Acknowledgments

We thank all respondents and stakeholders for their time and contributions toward making this work possible. The QCN Evaluation Group is: Nehla Djellouli, Kasonde Mwaba, Callie

Daniels-Howell, Tim Colbourn (UCL Institute for Global Health, UK), Kohenour Akter, Fatama Khatun, Mithun Sarker, Abdul Kuddus, Kishwar Azad (BADAS-PCP Bangladesh), Kondwani Mwandira, Albert Dube, Gladson Monjeza, Rachel Magaleta, Zabvuta Moffolo, Charles Makwenda (Parent and Child Health Initiative, Malawi), Mary Kinney, Fidele Mukinda (independent researchers, South Africa), Mike English (Oxford University), Yusra Shawar, Will Payne, Jeremy Shiffman (Johns Hopkins University, USA), Kathy Lubowa, Agnes Kyamulabi, Hilda Namakula, Gloria Seruwagi (Makerere University, Uganda), Anene Tesfa, Asebe Amenu, Theodros Getachew, Geremew Gonfa (Ethiopia Public Health Institute, Ethiopia), Seble Abreham, Tanya Marchant (LSHTM, UK)

## Author Contributions

**Conceptualization:** Kohenour Akter, Yusra Ribhi Shawar, Tim Colbourn.

**Data curation:** Nehla Djellouli.

**Formal analysis:** Kohenour Akter, Yusra Ribhi Shawar, Tim Colbourn.

**Funding acquisition:** Yusra Ribhi Shawar, Mike English, Jeremy Shiffman, Tim Colbourn.

**Investigation:** Kohenour Akter, Yusra Ribhi Shawar, Anene Tesfa, Callie Daniels Howell, Gloria Seruwagi, Agnes Kyamulabi, Albert Dube, Geremew Gonfa, Kasonde Mwaba, Mary Kinney, Mike English, Jeremy Shiffman, Nehla Djellouli, Tim Colbourn.

**Methodology:** Kohenour Akter, Yusra Ribhi Shawar, Anene Tesfa, Callie Daniels Howell, Gloria Seruwagi, Agnes Kyamulabi, Albert Dube, Geremew Gonfa, Kasonde Mwaba, Mary Kinney, Mike English, Jeremy Shiffman, Nehla Djellouli, Tim Colbourn.

**Project administration:** Callie Daniels Howell, Kasonde Mwaba, Nehla Djellouli, Tim Colbourn.

**Resources:** Kohenour Akter, Callie Daniels Howell, Nehla Djellouli, Tim Colbourn.

**Supervision:** Yusra Ribhi Shawar, Mike English, Jeremy Shiffman, Nehla Djellouli, Tim Colbourn.

**Validation:** Kohenour Akter, Yusra Ribhi Shawar, Anene Tesfa, Callie Daniels Howell, Gloria Seruwagi, Agnes Kyamulabi, Albert Dube, Geremew Gonfa, Kasonde Mwaba, Mary Kinney, Mike English, Jeremy Shiffman, Nehla Djellouli, Tim Colbourn.

**Visualization:** Kohenour Akter, Tim Colbourn.

**Writing – original draft:** Kohenour Akter, Yusra Ribhi Shawar, Tim Colbourn.

**Writing – review & editing:** Kohenour Akter, Yusra Ribhi Shawar, Anene Tesfa, Callie Daniels Howell, Gloria Seruwagi, Agnes Kyamulabi, Albert Dube, Geremew Gonfa, Kasonde Mwaba, Mary Kinney, Mike English, Jeremy Shiffman, Nehla Djellouli, Tim Colbourn.

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
