## [Decision Letter · Decision Letter 0]

9 May 2023

PGPH-D-23-00313

Influences on policy-formulation, decision-making, organisation and management for maternal, newborn and child health in Bangladesh, Ethiopia, Malawi and Uganda: the roles and legitimacy of a multi-country network

Dear Dr. Akter,

Thank you for submitting your manuscript to PLOS Global Public Health. After careful consideration, we feel that it has merit but does not fully meet PLOS Global Public Health’s publication criteria as it currently stands. Therefore, we invite you to submit a revised version of the manuscript that addresses the points raised during the review process.

EDITOR:

Thank you for your patience as we found suitable reviewers for this paper. The paper addresses legitimacy- and onwership-building for a multi-stakeholder, multi-country network for quality of care for maternal, newborn, and child health. The author team includes authors from each of the study countries, as well as the US, UK, and South Africa. The research is overall technically sound, though description could be improved by addressing reviewer comments. The research findings are relevant to other public health actors who may be utilizing a network approach to improve quality of care. Please respond to reviewer comments below (and see mark-up on manuscript from Reviewer 2).

We look forward to receiving your revised manuscript.

Kind regards,

Heather Haq, M.D., M.H.S.

Academic Editor

Journal Requirements:

1. Please include a complete copy of PLOS’ questionnaire on inclusivity in global research in your revised manuscript. Our policy for research in this area aims to improve transparency in the reporting of research performed outside of researchers’ own country or community. The policy applies to researchers who have travelled to a different country to conduct research, research with Indigenous populations or their lands, and research on cultural artefacts. The questionnaire can also be requested at the journal’s discretion for any other submissions, even if these conditions are not met.  Please find more information on the policy and a link to download a blank copy of the questionnaire here: https://journals.plos.org/plosone/s/best-practices-in-research-reporting. Please upload a completed version of your questionnaire as Supporting Information when you resubmit your manuscript.”

Additional Editor Comments (if provided):

Reviewers' comments:

Reviewer's Responses to Questions

**Comments to the Author**

1. Does this manuscript meet PLOS Global Public Health’s publication criteria? Is the manuscript technically sound, and do the data support the conclusions? The manuscript must describe methodologically and ethically rigorous research with conclusions that are appropriately drawn based on the data presented.

Reviewer #1: Yes

Reviewer #2: Yes

2. Has the statistical analysis been performed appropriately and rigorously?

Reviewer #1: N/A

Reviewer #2: N/A

3. Have the authors made all data underlying the findings in their manuscript fully available (please refer to the Data Availability Statement at the start of the manuscript PDF file)?

Reviewer #1: No

Reviewer #2: No

4. Is the manuscript presented in an intelligible fashion and written in standard English?

Reviewer #1: Yes

Reviewer #2: Yes

5. Review Comments to the Author

Reviewer #1: I applaud the authors for this carefully prepared paper. I am happy to recommend minor revisions to this paper, as it is already well thought out and written.

The premise that QCN needs legitimacy to work is one that is sound. As the authors write, “for the network to work as intended the government of each member country must buy into the idea and spend time and resource on network activities and coordination” (p. 3, line 57). To study these interactions, the authors draw on a wide range of perspectives on legitimacy to explain different degrees of buy-in to QCN by national and local policymakers in the four countries they study (of Bangladesh, Ethiopia, Malawi, and Uganada, as well as at the global level). The analysis focuses on “political, normative and cognitive interactions between institutions involved in QCN in each country” (p. 9, line 146).

I am writing here from a different disciplinary background as the authors, so please take these literature suggestions with a grain of salt and use as you see fit—we also don’t typically talk about legitimacy, but we do think in terms of trust. Here work from the sociology of global health may be useful in reframing some of the questions that the authors have (Harris and White 2019; Robinson 2017). These approaches generally, like the authors, take into account the institutional contexts and histories of the countries under study, but attempts to link through sequencing and looking at changes in “field configurations”, how actors build on previous meanings attributed to certain public health interventions (e.g. in Robinson, HIV/AIDS interventions are mapped onto the family planning field that preceded it). Drawing from this literature can also help link the material provided in the background sections to the empirical findings of the “present”. Potentially relevant also is the emerging literature from economic sociology on network failures, which also sees trust and legitimacy (as well as competence and opportunism) as key to the success of networks such as QCN, as well as other public-private partnerships, international organizations, supply chains etc. (Schrank and Whitford 2011), as well as with works in the sociology of development that looks at “pockets of efficiency” within large bureaucracies and the “reactive diffusion” of global networks with values, norms, priorities, etc. (Chorev 2019; McDonnell 2020).

The comparative framework, research design, and methods is also sound. One of the methodological problems that other studies of this sort have is to assume that “legitimacy” is a thing that is out there to be measured and captured—that respondents either believe there is legitimacy, or not, or possess it to a certain degree that can be quantified. Instead, the authors focus on social processes that construct legitimacy. The authors are able to use iterative interviews from 2019 to 2022 to trace these processes. The methods are also described in Supplement 2, but it would be helpful to give a quick gesture to the number of interviews, length of interviews, etc. in the main text so readers can more easily locate this information. The supplements are also very detailed, and I applaud the authors for the care they took to document the data and methods they employed. The author’s request to withhold interview data should also be granted—as is the norm for qualitative interviews of this sort of potentially identifiable respondents.

The analysis and results sections are presented first by providing necessary background into the four cases, and then looking at the political, normative, and cognitive interactions that create legitimacy. Illustrative quotes are presented to highlight these interactions at work. And the discussion section goes over these more broadly and summarizes the presence/absence of these interactions in the four cases. One question I have with this section is that it seems like the authors are taking an “additive approach” to legitimacy: Legitimacy is calculated or qualitatively described by adding up the presence of all these different types of interactions. But are there types of interactions that lead to less legitimacy? For instance, if actors learn of information (cognitive interactions) that is interpreted as negative or bad, does this detract from legitimacy of QCN? Some clarification would be helpful here.

The conclusion that “we find evidence supporting the legitimacy of QCN from the perspective of country governments, further work and time are required for governments to own and embed the work of QCN in routine care” is also sound. But it would be helpful for the authors to develop this point a little bit further and develop specific recommendations on how QCN can be incorporated in routine care. Here the authors should also address how legitimacy is different when implemented at the micro-level, in clinical settings, etc.

Minor points:

- On p. 2, line 32 (and subsequent uses of this phrase): “We developed a framework drawing on three frameworks”, perhaps change one of the frameworks to perspectives? Or some other synonym.

- On p. 28, line 561: “We found political interactions to be good in all four case study countries…”, perhaps replace “good” here with something referring to the frequency and quality of these interactions? “Good” sounds like you are judging the outcomes of these interactions here.

References:

Chorev, Nitsan. 2019. Give and Take: Developmental Foreign Aid and the Pharmaceutical Industry in East Africa. Princeton University Press. https://press.princeton.edu/books/hardcover/9780691197852/give-and-take.

Harris, Joseph, and Alexandre White. 2019. “The Sociology of Global Health: A Literature Review.” Sociology of Development 5 (1): 9–30. https://doi.org/10.1525/sod.2019.5.1.9.

McDonnell, Erin Metz. 2020. Patchwork Leviathan: Pockets of Bureaucratic Effectiveness in Developing States. Princeton University Press. https://press.princeton.edu/books/hardcover/9780691197357/patchwork-leviathan.

Robinson, Rachel Sullivan. 2017. Intimate Interventions in Global Health: Family Planning and HIV Prevention in Sub-Saharan Africa. Cambridge: Cambridge University Press. https://doi.org/10.1017/9781316117033.

Schrank, Andrew, and Josh Whitford. 2011. “The Anatomy of Network Failure.” Sociological Theory 29 (3): 151–77. https://doi.org/10.1111/j.1467-9558.2011.01392.x.

Reviewer #2: This manuscript assessing the legitimacy, ownership and embedded-ness acquired by the Quality Care Network in four separate countries and as an international entity is an ambitious and important project. My impression is that this could contribute to the global knowledge base through publication, however, some improvements should be made first. Details can be found in the uploaded document. A broad summary of the areas chosen for improvement: first, an overview of the QCN structure and activities--what are the key agreements that make it happen, what are the roles and responsibilities of participants? What motivates participation in the QCN? Similarly, summary statements about what was found in the other papers about QCN and how they relate to this analysis--which of the case study countries were found to have effective QCN for example, and did that finding correspond to high legitimacy in this analysis? Another suggestion relates to greater attention to the working definitions of the key terms used in the analysis, beginning with "legitimacy." The chosen characteristic, an entity's ability "to exert power," does not seem wholly fitting. The figure depicting knowledge exchange with the differently weighted arrows could benefit from a re-work with more detailed explanation. Overall, more examples to support the statements would support the piece and its conclusions better. It seems the authors have created a new framework by adapting three for this application--another figure expounding the adapted framework would support the conclusion that the framework could be used in other applications and could help readers digest the study. Finally, another round of proofreading should be done. Some of the corrections are noted in the attached file. Again, this work is appreciated as an uncommon attempt to make sense of how multi-country networks convened by global level actors achieve influence or fail to do so--its publication would be a contribution to global health practice. Best of luck to the authors for a successful round of edits.

6. PLOS authors have the option to publish the peer review history of their article (what does this mean?). If published, this will include your full peer review and any attached files.

**Do you want your identity to be public for this peer review?** For information about this choice, including consent withdrawal, please see our Privacy Policy.

Reviewer #1: No

Reviewer #2: No

---

## [Decision Letter · Decision Letter 1]

16 Oct 2023

Influences on policy-formulation, decision-making, organisation and management for maternal, newborn and child health in Bangladesh, Ethiopia, Malawi and Uganda: the roles and legitimacy of a multi-country network

PGPH-D-23-00313R1

Dear Ms Akter,

We are pleased to inform you that your manuscript 'Influences on policy-formulation, decision-making, organisation and management for maternal, newborn and child health in Bangladesh, Ethiopia, Malawi and Uganda: the roles and legitimacy of a multi-country network' has been provisionally accepted for publication in PLOS Global Public Health.

Best regards,

Heather Haq, M.D., M.H.S.

Academic Editor

Reviewer Comments (if any, and for reference):

Reviewer's Responses to Questions

**Comments to the Author**

1. If the authors have adequately addressed your comments raised in a previous round of review and you feel that this manuscript is now acceptable for publication, you may indicate that here to bypass the “Comments to the Author” section, enter your conflict of interest statement in the “Confidential to Editor” section, and submit your "Accept" recommendation.

Reviewer #1: All comments have been addressed

Reviewer #2: All comments have been addressed

2. Does this manuscript meet PLOS Global Public Health’s publication criteria? Is the manuscript technically sound, and do the data support the conclusions? The manuscript must describe methodologically and ethically rigorous research with conclusions that are appropriately drawn based on the data presented.

Reviewer #1: Yes

Reviewer #2: Yes

3. Has the statistical analysis been performed appropriately and rigorously?

Reviewer #1: Yes

Reviewer #2: N/A

4. Have the authors made all data underlying the findings in their manuscript fully available (please refer to the Data Availability Statement at the start of the manuscript PDF file)?

Reviewer #1: No

Reviewer #2: Yes

5. Is the manuscript presented in an intelligible fashion and written in standard English?

Reviewer #1: Yes

Reviewer #2: Yes

6. Review Comments to the Author

Reviewer #1: Thank you for the opportunity to re-review this manuscript and read the response letter. I applaud the authors for their engagement with reviewer comments and the incorporation of our suggestions. I am satisfied with the changes that the authors have made, and they have sufficiently addressed my concerns. The new language added on the limitations of methods, and their engagement with the broad literature in sociology on institutions and context in global health is laudable. I am happy to recommend an acceptance of this piece!

Reviewer #2: I am satisfied with the revised manuscript. It is now more clear and detailed and therefore has more potential to be a powerful influence in global health. Congratulations and thanks to the authors for undertaking and sharing this work.

7. PLOS authors have the option to publish the peer review history of their article (what does this mean?). If published, this will include your full peer review and any attached files.

**Do you want your identity to be public for this peer review?** For information about this choice, including consent withdrawal, please see our Privacy Policy.

Reviewer #1: No

Reviewer #2: No
